# Bilingual translations of intensifiers in Dong-A Ilbo's news about China: A corpus-based discourse analysis approach

Quan Jiuding [ORCID] *

Shanghai Ocean University, Shanghai, China

* 1459335419@qq.com

## Abstract

Drawing on the framework of Ideological Square Model, this paper carries out a corpus-based analysis of the ways in which group relations and the image of China are re-shaped in the English translation (ET) and the Chinese translation (CT) of Korean news discourse about China with intensifiers as an entry of inquiry. The results show that (1) There is a statistically significant difference regarding faithfulness between the ET and the CT of intensifiers ($\chi^2$ = 38.11>3.84, $p$<0.05), with the ET having more translation shifts. Additionally, the Chi-square test of a 3x3 contingency table (T = 49.77>5.99, P<0.05) indicates that there is a difference between the distribution of translation shifts at the 3 levels between ET and CT. (2) The ET of Korean news about China aligns more closely with van Dijk's ideological square model, while CT violates. It is argued that South Korean media have ideological factors to consider when it comes to Chinese readers accepting CT. (3) There is an alteration of the ideological square model in the target texts. The results indicate that in the translation of a target language, the translation propensity for relevant topics of the target country in which the language is spoken obeys the rules of "Us", the ingroup, but not "them", the outgroup.

## Introduction

Translation shift has long been a major concern in translation studies. Various aspects have also provided deeper insight into translation shifts, including linguistic elements, translation universals, and even social and psychological factors [1–3]. In recent years, ideological factors have also become more prominent in translation shift research. Li and Pan [4] analyze China's image reshaping through Chinese political discourse. Gu and Tipton [5] explore government-affiliated interpreters' mediation of Beijing's discourse on different levels using self-referential terms.

Translation shifts and ideology are most studied in monolingual translation texts [6,7]; translation texts of different historical periods are examined for the evolution of ideology over time [8], as well as comparing translations of a source text into multiple versions in order to identify ideologies expressed by various translators [9]. However, there have been few studies examining the role of ideology in multilingual translation.

**Data Availability Statement:** All relevant data are within the paper and its Supporting Information files.

**Funding:** The author(s) received no specific funding for this work.

**Competing interests:** The authors have declared that no competing interests exist.

Since the internationalisation and localisation of products such as software and websites already strengthened the Globalisation, Internationalisation, Localisation and Translation (GILT) industry [10], translation has therefore become a multilingual research topic that has given rise to a flourishing field of translation studies in recent years [11]. Restrictions and capabilities of dubbing and subtitles were discussed when dealing with multilingualism [12]. Shen L. [13] conducts a 50-year (1970–2019) comparison of referential explicitness between an 11,721,608-token corpus of English-translated diplomatic discourse from 56 languages and an 11,113,036-token corpus of English original diplomatic discourse extracted from the United Nations General Debate Corpus (UNGDC) with the Multi-dimensional Analysis (MDA) framework. A multilingual corpus of selected English children's books translated into German, Greek, Korean, Spanish, and Arabic is used to illustrate how ideological manipulation of originals leads to shifts in translation in different languages [14].

Therefore, this study aims to explore ideological manipulation in multilingual translation. Today, multilingual translation texts are most often found in the news media due to globalization. In the digital era, news discourse interpretation plays an increasingly significant role in constructing ideologies for global audiences. Kim K H. [15] argues that mass media play a central role in circulating and disseminating ideas and thus offer an interesting arena for analysis of how media discourses are constructed, disseminated, and mediated via translation from Korean into English. In news media discourse, translation shifts are also frequently observed. As translation of news discourse into another language often requires trans-editing, which means many translation shifts occur in the process, and news discourse is often imbued with ideologies, the genre of news has attracted much attention in translation studies [16,17].

Geographically, South Korea is an important neighbor of China in East Asia. Its high level of public diplomacy capability and its large circulation of news discourse in the Asian cultural circle also produces a profound influence on the world at large. Terry, who currently works at the Wilson Center, wrote an article for *Foreign Affairs* where she noted that"South Korea has now become a global soft-power juggernaut"[18]. Moreover, South Korea belongs to the Confucian culture circle, but also maintains close ties with both China and the United States as well. This can be evidenced by the former South Korean President Moon Jae-in's pursuit of balanced diplomacy with the U.S. and China, which is perceived as more neutral in its diplomatic approach. Such diplomatic positioning is also reflected in Korean news discourse. The ways in which the diplomatic positioning are translated into other languages, particularly Chinese and English, become an interesting topic to investigate.

The Dong-A Ilbo has been a representative of Korean news media. According to the website introduction, it has been published since 1920, covering more than 1.2 million daily issues and being read primarily by opinion leaders. The motto of this publication is "For the people, democracy, and culture". It has also partnered with *The New York Times*, *The Asahi Shimbun*, and China's *The People's Daily* [19]. As the representative news media of Korea's elite, Dong-A Ilbo may have provided insight into Korea's international political position.

This paper aims to examine whether or not there is a significant difference between English translation(ET) and Chinese translation(CT) of Korean news discourses, and explores whether the translation shift is influenced by ideological factors. It is therefore interesting and would offer insights into translation studies and journalism as well by examining the different features of ET and CT of Dong-A Ilbo' s discourse about China and the ways in which a representative news outlet in South Korea, a neighboring country of China, reshaped China's image and group relations. Furthermore, this study also attempts to reveal the mechanisms of reproduction in the multilingual translations in news media.

## Ideological square model

Lefevere A. [20] argues that translation can be regarded as a form of rewriting, and sees translation as an act carried out under the influence of particular categories and norms constituent to systems in a society. The most important of these are patronage, ideology, poetics, and 'the universe of discourse'. Venuti L. [21] defines the translation texts as a kind of corpus with the ideological qualification to assume a role of performing a function in an institution. Recent years have witnessed the increasing attention in translation studies to the contextual analysis of translation products, particularly the ideological factors that influence the production of translation. In this way, the scope of the research has been broadened and more theoretical models have been introduced to the analysis of translation and interpretation. Accordingly, Munday J. and Zhang M. [22] have contributed to the study of the translators' ideology through the use of appraisal theory developed by Martin and White. Furthermore, van Dijk proposes a general theory of ideology and its reproduction by discourse, in which dominant ideologies are reproduced, with racism, sexism, classism, or neoliberalism as examples [23]. This approach to ideology presented here is seen as part of Critical Discourse Studies(CDS), which examines the ways of social power abuse embedded in discourse [24,25].

According to van Dijk, ideological discourse usually exhibits the polarized structure of underlying attitudes and ideologies that emphasizes the positive tendency of "Us", the ingroup, and the negative tendency of "Them", the outgroup [23]. All levels of discourse would be affected by this polarization. Ideologies are generally presented by text, talk, and other forms of communication, while ideological discourse structure facilitates the formation of ideological models, attitudes, and ideologies. It is therefore common in ideological discourse to emphasize the negative qualities of outgroups and positive qualities of ingroups, while ignoring, suppressing, or mitigating our negative qualities and their positive ones [23].

Ideologies are formed through shared attitudes about social issues that are relevant to the group and its reproduction [26]. As a result, attitudes influence group members' personal mental models about specific events and actions. These mental models again shape actual social practices, such as the production and comprehension of discourse, and even the discourse of translation.

Specifically in the theoretical framework, van Dijk argues that there are several key aspects to ideological discourse semantics, i.e., topics, propositions, modalities, local coherence, implications, presuppositions, actor descriptions, disclaimers, metaphors, and the level and granularity of events and action descriptions. However, news translation is most influenced by the manipulation of topics and the granularity of event descriptions [23]. First, discourse topics can be influenced by underlying attitudes and ideologies. In the usual polarized structure of ideological discourse, it can be expected largely negative topics are presented about Them and neutral or positive ones about Us. Similarly, at each level of description, a speaker may give many or few component descriptions or actions or events. Again, in ideological discourse, such variation may well be biased against others. It is therefore likely that negative actions or attributes of the Others are usually described in a way that is not only global, but also at very specific levels, and often with greater granularity, as a semantic-rhetorical means of emphasizing the point. In general, "Our" negative actions will be described at very general or abstract levels, and not in great detail.

## Methodology

The corpus-based discourse analysis (CDA) utilizes methodological and computational innovations that allow scholars to ask novel, frequency-based research questions on existing linguistic phenomena across many speaking and writing contexts [27]. In Translation Studies,

**Table 1. The Set of Intensifiers(I) and Downtoners(D).**

| Term | Type | Factor | Term | Type | Factor |
|---|---|---|---|---|---|
| Very | I | 1.25 | too | I | 1.25 |
| Highly | I | 1.75 | absolutely | I | 1.75 |
| Really | D | 0.75 | completely | I | 1.5 |
| More | D | 0.50 | most | I | 1.75 |
| Always | I | 1.5 | mostly | D | 0.75 |
| Many | I | 1.25 | pretty | D | 0.75 |
| Fully | I | 1.5 | somewhat | D | 0.5 |
| So | I | 1.25 | slightly | D | 0.75 |
| Lot | I | 1.5 | extraordinarily | I | 2.0 |
| Full | I | 1.5 | ever | I | 1.75 |
| Extremely | I | 2.0 | still | D | 0.75 |
| totally | I | 1.25 | just | D | 0.75 |
| little | D | 0.25 | barely | D | 0.25 |
| Slightly | D | 0.5 | hardly | D | 0.25 |

corpora have provided a basis for empirical descriptive research [28–32]. As Baker [33] argues, corpora can be used to observe features that have been noted in translated texts in a systematic way. Accordingly, the translation shift is evident in frequency data, and can be analyzed in terms of the topics and semantic prosody of the collocation words [34–36].

The translation shift of intensifiers is most evident in this multilingual corpus, which is also supported by subsequent data analysis. As a word class, intensifiers are the most likely to undergo semantic shifts [37], and they are also likely to influence listeners' reception of what is said [38]. The classification of intensifiers in the appraisal system is used widely in linguistic and translation studies. Quirk R. [39] categorizes amplifiers as maximizers and boosters. As stated by Biber D, Johansson S and Leech G N. [40], these can be used to indicate that the extent or degree is greater or less than usual compared to something else in the surrounding discourse and they occur both as adverbs and modifiers. Downtoners are defined as degree adverbs that scale down the effect of the modified item [41]. Downtoners are classified into four classes [41,42]: approximators, which express an approximation to the force of verb or predicate by indicating a state of nearly, but not completely or not quite, e.g nearly, practically, almost; 2) compromisers, which have a slight lowering effect, e.g. kind of, more or less, rather, quite; 3) diminishers, which scale down the force of the utterance, with the meaning of 'to a small extent', e.g. a little, somewhat, slightly; and 4) minimizers, which tend to have the meaning'(not) to any extent', e.g., barely, little, hardly, in the least). In order to better visualize the different levels of intensifiers, the Aspect-based semantic analysis of intensifiers is applied in this study for reference as shown in Table 1 [43].

## Corpus design

This corpus is a trilingual one consisting of two parallel corpora, i.e., the Korean-English parallel corpus and the Korean-Chinese parallel corpus, containing international news about China reported by *Dong-A Ilbo* [44] from 10 May 2017 to 9 May 2022 in bilingual form during Mun Jae-in's tenure. All the documents and their translations can be downloaded from the Dong-A Ilbo website, using the extraction word "China".The choice of this particular time frame for our research is driven by the intention to minimize the influence of government policy changes on news media's translation strategies. Government policies can significantly impact how news is reported and translated, and these policies evolve over time. By focusing on a

specific time frame, this study aims to capture a more stable and consistent period in news reporting, reducing the confounding variables introduced by frequent policy changes. The selection of our sample size is guided by the need to achieve statistical reliability while still maintaining a manageable dataset. A larger sample size can provide more robust insights into translation strategies. However, considering the time and resources available for our research, this study have opted for a sample size that strikes a balance between statistical significance and practicality. This ensures that our analysis remains feasible within the constraints of the research scope, while still allowing us to draw meaningful conclusions about news media translation strategies during the specified time frame. In total, 188 files were collected, totaling 335,589 characters, including 143,962 Korean characters, 126,963 Chinese characters, and 64,664 English words.

It is noted that the decision to view the media as a cohesive entity, rather than those roles of publishers, editors, and translators, is based on recognizing the complex and interconnected nature of news production and translation processes. In many media organizations, these roles work closely together as integrated systems to create news content. This integrated perspective acknowledges that translation decisions in news media rarely happen in isolation but rather result from dynamic interactions and negotiations involving multiple stakeholders. By treating the media as a unified entity, our research aims to offer a more comprehensive view of the translation strategies used in news reporting. Understanding how translation choices are made within the broader context of media practices allows us to explore the various factors that influence these decisions [45].

## Data collection

First, using Tmxmall to align the data at sentence level in the two corpora and observe the parallel concordances, this paper finds that there is no obvious shift in the translation of topics, propositions, modalities, local coherent and actor descriptions, but a significant change in the granularity and level of event and action descriptions. Consequently, intensifiers are chosen as the object of research, as they are the most representative linguistic element in terms of level and granularity.

Second, the 28 adverbs of degree used as intensifiers were selected from the top 100 adverbs in the Sejong Corpus, which is a large-scale Korean language corpus developed by the National Institute of Korean Language in South Korea [46], with 72 intensifiers that are not present in the two parallel corpora not included for further anlaysis. This list of 28 intensifiers includes 24 amplifiers and 4 downtoners, covering maximizers such as '제일' [best], '확실히' [definitely], '완전히' [totally], '전적으로' [entirely], '충분히' [fully], '철저히' [thoroughly],'가장' [most], '진짜' [truly],'전혀' [absolutely], '결코' [never], '반드시' [certainly]; Boosters as '고도로' [highly], '너무' [too], '매우' [very], '대단히' [greatly], '크게' [largely], '상당히' [significantly], '대대적으로' [in a big[great, large] way], '강하게' [strongly], '정말' [really], '잘' [nicely], '강력하게 ' [hard],'거세게' [fiercelly], '견고하게' [stably],'아주' [very], Approximators as '기본적으로' [basically]; Compromisers as '상대적으로' [relatively]; Diminishers as '다소' [somewhat]; Minimizers as '거의' [hardly]. As a final result, 169 concordances of intensifiers of degree are retrieved from the Korean news source.

Third, Key Words in Context (KWIC) in Tmxmall are used to extract those 169 intensifiers in two sub-corpora with the information tags, i.e., <maximizers>, <approximators>, <compromisers> and <minimizers>. Then in terms of the types of intensifiers and translation corresponding words, the changing level of intensifiers are tagged in three scales, i.e., <unchanged>, <up-scale>, or <down-scale>. Marking the variations in intensifiers was determined rigorously based on whether corresponding intensifiers existed in the Korean-

**Table 2. Topics and semantic prosody of collocations related to "China".**

| Topics of collocation words | Semantic prosody of collocation words | number |
|---|---|---|
| China-GR | positive | 26 |
| | negative | 33 |
| | neutral | 2 |
| China | positive | 22 |
| | negative | 32 |
| | neutral | 16 |

Chinese and Korean-English dictionaries of the Naver Dictionary (comprising a range of dictionaries including Oxford, Longman, and Goryeo Hanja). If there were no corresponding words, the changes were tagged based on Quirk's classification and the sentiment analysis data on intensifiers in Table 1. Subsequently,translation strategies were tagged as <literal translation>, <free translation> and <zero translation>. In this study, literal translations of intensifiers are used to describe the use of corresponding words existed in dictionaries. Free translation refers not only to the absence of the corresponding word but also to changes at the intensifier level or in semantic meaning. In the zero translation, intensifiers have been omitted directly by the translators.

Furthermore, statistical analysis of translation strategies was conducted based on whether there were changes in intensifiers. If the corresponding words was present in dictionaries, it was categorized as a literal translation; if not, it was categorized as a free translation. In cases of omission in translation, it was classified as a zero translation. Furthermore, for a more thorough analysis, collocations that describe China were tagged into three categories: <national image> (referring to China's policies), <group relations> (referring to China's interaction with the United States), and <irrelevant> (referring to natural resource words such as dust, ore, and weather). Semantic prosody is considered to be the overall discourse function of a 'unit of meaning' in a text [47]. It refers to the inherent or associated evaluative or affective meaning that a word or phrase carries, which may influence the overall interpretation or emotional impact of a text or discourse. After a close reading of each parallel concordance lines of these intensifiers in 2 target translation corpus, collocation words are also characterized by their semantic prosody, which can be divided into 3 categories: <positive>, <negative>, and <neutral>. Through the analysis and annotation of the source text, it has allowed us to analyze the collocational tendencies and semantic prosody of the source text based on data, as shown in Table 2. Expanding upon these indicators, the patterns of variation in intensifiers can be clearly evident.

Accordingly, we divide intensifies into nine points of level change, as shown in Table 3. A noteworthy point is that, when zero translation occurs, the omission of amplifiers causes the <down-scale>, whereas the omission of downtoners causes the <up-scale>. For example,

**Table 3. Distribution of 9 level-changing points.**

| Chinese Translation | English Translation | | | Total |
|---|---|---|---|---|
| | Unchanged | Up-scale | Down-scale | |
| Unchange | (A) 99 | (B) 16 | (C) 40 | 155 |
| Up-scale | (D) 1 | (E) 1 | (F) 0 | 2 |
| Down-scale | (G) 6 | (H) 2 | (I) 4 | 12 |
| Total | 106 | 19 | 44 | 169 |

when the amplifier '매우' [very] co-occurs with words indicating positive China's image, the up-scale/down-scale of amplifiers' level makes the up-scale/down-scale of China's image, while the zero translation causes the down-scale of it. However, when the downtoner '다소' [little] co-occurs with words indicating positive China's image, the up-scale/down-scale of downtoners makes the up-scale/down-scale of China's image, while the zero translation of downtoners makes the up-scale of the positive prosody. Furthermore, although almost all the zero translation of intensifiers in translation causes the level-changing, free translations are not. And all these factors are taken into account when tagging the level changing labels.

Last but not least, it should be noted that this study uses the Chi-square statistical method to address several issues. This research aims to identify statistically significant differences between ET and CT in terms of faithfulness when translating intensifiers. Chi-square tests are reliable statistical tools for comparing categorical data and are particularly suitable for testing whether the observed differences between translations are statistically significant. In addition, the study involves analyzing translation shifts across three levels, which is a situation that is well-suited to chi-square testing with 3x3 contingency tables [48–50]. By systematically examining the distribution of translation shifts, this study gains a more comprehensive understanding of the differences between ET and CT. Furthermore, the objective of this study is to examine whether the translations were in accordance with van Dijk's ideological square model. The chi-square test allows us to quantitatively assess whether the translations adhered to or deviated from this model, providing valuable insights into the ideological factors at work. Finally, this study aims to investigate alterations in the ideological square model within the translation texts. Chi-square analysis allows us to investigate whether translations exhibited a tendency to align with "Us" (the ingroup) or "them" (the outgroup), providing insight into the dynamics of ideological shifts in translation. Accordingly, the choice of chi-square statistical analysis is driven by the specific objectives of this study, which included quantitative and qualitative analyses of translation shifts, alignment with ideological models, and shifts in ideological perspectives.

## Results

### Differences between ET and CT of intensifiers

Based on the results of the paired Chi-square test in Table 4 ($\chi^2$ = 38.11>3.84, $p$<0.05), there is a gap in the overall distribution of faithfulness between Chinese and English translations, with a lower level of fidelity in English translations than in Chinese translations. Thus, it is necessary to conduct a detailed study of translation shifts. The source text of Korean news has two versions of translation (ET and CT), and the translation of intensifiers generates three types of translation effects: unchanged, up-scale, and down-scale. Unchanged is achieved through literal translation and some free translations which have no semantic shift according to the categories of intensifiers and semantic analysis as shown in Table 1, while level-changing is

**Table 4. Fidelity distribution of intensifiers between ET and CT.**

| Chinese Translation | English Translation | | Total |
|---|---|---|---|
| | Literal translation | Free translation+ Zero translation | |
| Literal translation | 99 | 56 | 155 |
| Free translation+ Zero translation | 7 | 7 | 14 |
| Total | 106 | 63 | 169 |

generally achieved through zero translation and free translation which has shift of the categories of intensifiers or sentimental factors. A categorical variable obtained through counting the frequency of inhomogeneous data is included in the processed and listed statistics (Table 3) as enumeration data. The Chi-square test is applied to examine the contingency table of three types of translation effects in two target texts to determine whether intensifiers have the same translation effect on two target texts [50]. The following is the calculation process.

**Hypotheses**:

H0: The distribution of 3 translation effects probability in two target texts is the same.

H:1:The distribution of 3 translation effects probability in two target texts is different.

Level of significance α: 0.05

**Calculate test statistic**:

$$T = \frac{k-1}{k} \sum_{i=1}^{k} \frac{(n_i - m_i)^2}{n_i + m_i - 2A_{ii}}$$

$$= \frac{3-1}{3} \times \left[ \frac{(155-106)^2}{155+106-2\times99} + \frac{(2-19)^2}{2+19-2\times1} + \frac{(12-44)^2}{12+44-2\times4} \right] = 49.77$$

**Decision**:

If the $p$-value is smaller than α, we reject H$_0$ at the significance level α.

If the $p$-value is bigger than or equal α, we fail to reject H$_0$ at the significance level α. According to the Chi-Square test ($\chi^2$), $\chi^2_{0.05,2}$ = 5.99, T(49.77)>5.99 based on the degree of freedom is 2 (3variables-1), and P<0.05. Therefore, H$_0$ is rejected and H$_1$ is accepted when α = 0.05.

The result shows that there is a significant difference in the distribution of 3 translation effects' probability in two target texts. And shifts usually occur in the topics of group relation and national image(≈97%) rather than the irrelevant words which mainly consists of natural resources words, such as ore, dust, and weather (2%). According to van Dijk's, an editorial in a newspaper displays the ideology of the press [51]. Considering ideology is a relevant property of participants, it also influences discourse variation, not just in terms of its content or meaning, but also in terms of its expression. Considering the formal nature of the news discourse in Dong-A Ilbo, the significant difference in translation effects and the level-changing of intensifiers in the two target texts indicate the manipulation of the translation texts due to the news outlet's ideology. Hence, two target texts will be examined for their translation shift of intensifiers towards topics and semantic prosody of collocations.

## Evaluation of source text

As those intensifiers appear as nodes in the concordance lines, the most common collocation words are words indicating national image and group relation. Hunston noted that prosodic collocations reveal "the speaker or writer's attitude or stance towards viewpoint or feelings about the entities and propositions that he or she is talking about" [52]. Futhermore, the topics and semantic prosody of collocation words can be used to determine the ideology behind the translator's manipulation of intensifiers. Therefore, the evaluation of trilingual discourse will be based on this model as Table 5.

**Example 1**:

ST-kor: 런민일보는 "메르켈이 재임하는 동안 중국과 독일 관계는 물론이고 중국과 유럽연합(EU)의 관계도 매우(very) 돈독해졌다"고 평했다

TT-en: The People's Daily wrote that the China-Germany relations and the China-EU relations have been strengthened during Merkel's time in office

**Table 5. Analysis of discourses.**

| Collocation Words | | Node | Translation Shift of Node | |
|---|---|---|---|---|
| Topics | Semantic prosody | Intensifiers | Translation strategy | level-changing of Intensifiers |
| National image of China | Positive | Amplifiers | Literal translation | Up-scale |
| Group relation about China | Negative | Downtoners | Free translation | Down-scale |
| | Neutral | | Zero translation | Unchanged |

TT-cn:《人民日报》评价说:"默克尔在任期间,不仅是中国和德国的关系,中国和欧盟的关系也变得非常(very) 深厚"

**Example 2**:

ST-kor: 대만에 대해서는 '분할할 수 없는 중국 영토의 일부'라고 전제한 뒤 "조국의 평화통일을 추진하는 것은 중국 정부가 견지해온 방침으로, 미국 일부 세력이 대만 독립 세력을 지원하는 것은 매우(very) 잘못되고 위험한 것"이라고 경고했다

TT-en: Defining Taiwan as an undividable part of Chinese territory, he argued that China's pursuit of a nationwide peaceful unification has been maintained for so long, warning that it is a wrongful and risky act for some forces in the United States to support the independence of Taiwan

TT-cn: 他就台湾问题表示:"台湾是中国领土不可分割的一部分"他同时警告说:"推进祖国和平统一是中国政府一直坚持的方针,美国部分势力支援台独势力是非常(very)错误和危险的"

In Example1, the topic "China-Germany relations" and "China-EU relations" can be identified as a group relation, and the verb "strengthen" can be identified as a kind of positive activity. Considering its positive collocation in its context, the intensifier "very", an amplifier to raise the force of graduation, shows a positive translation prosody in the discourse. Moreover, judgment resources and appreciation resources exist in attitude resources of behavior, and capacity in other countries as Example 2. "China criticizes the US strategy" speaks to a negative group relationship between China and the US, and the collocation words are two judgement resources "risky" and "wrong". Thus, the semantic prosody is negative since from its extended unit of meaning. Likewise, the intensifier "very", an amplifier to raise the force of graduation, raises the level of negative prosody in the discourse.

This shift in intensification can be seen as a shift in meaning of the gradation strength, as intensification is a component of graduation. Therefore, the amplifier applied zero translation in these two examples indicates the down-scale translation propensity.

## Translation shift caused by the level-changing of intensifiers

At the level-changing point A (Table 3), there is no translation shift both in ET and CT, but the proportion of negative topics (NT) is the highest. In China's group relations, the frequency of collocation words is ranked as negative, positive, and neutral, and there are approximately 27% more negatives than positives. Considering China's national image (CNI), the frequency ranking of collocation words is negative, neutral, and positive, and the negative images of China (NIC) are almost 2.22 times those of PIC. Even without translation shift, the ratio of NIC to negative group relations(NGR) is the highest.

At level-changing point B, intensifier levels upscale in ET while those in CT remain unchanged. The number of positive group relations (PGR) for China is 0, whereas the number for NIC is 3.5 times that of PIC. In ET, the unilateral up-scale of intensifiers is mainly seen with the collocation of NIC. As in example 3, "largely" has been changed to "radically" with a up-sclae in graduation in the ET when describing the NIC. Several of the up-scale of positive

images of China (PIC) do not conform to the ideological square model at the lexical level. Analysis of this data, however, reveals that most of the level-changing are related to China's development goals or the government's power. Example 4 illustrates how a downtoner "basically" are dealt with through zero translation in the ET, which ultimately elevates the degree of Chinese socialist modernization from a vague degree to a clearly higher degree. For Example 5, zero translation of the downtoner "nearly" greatly strengthens the power of the government in ET.

Thus, it can be concluded that ET is strengthening NIC and NGR, and each upscale in PIC is for the purpose of strengthening the Chinese government's power.

**Example 3**:

ST-kor: 중 활동을 한 홍콩 시민을 최대 무기징역에 처할 수 있도록 한 홍콩 국가보안법 시행 등으로 홍콩의 사회 환경이 보안법 시행 전과 크게 (largely(0.75)) 달라진 여파 때문이라는 분석이 나오고 있다

TT-en: Since the Hong Kong Security Act, which allowed the state to sentence protestors in opposition to mainland china to life imprisonment at maximum, entered into force, Hong Kong's social environment has radically(1.5) changed, and this may have contributed to the large exodus of students and teachers.

TT-cn: 有分析认为,这是因为香港实施对进行反中活动的香港市民最高可判处无期徒刑的《国家保安法》等,香港的社会环境与《国家保安法》实施前发生了很大的 (largely(0.75)) 变化。

**Example 4**:

ST-kor: 단계로 2020년까지 이룬 전면적 샤오캉 모든 국민이 풍족하게 생활하는 것) 사회의 기초 위에서 2025년까지 사회주의 현대화를 기본적으로(basically(0.75)) 실현하고,

TT-en: During the first phase of development, Xi aims at prosperity of every citizen until 2020 and modernization of socialism until 2025

TT-cn: 第一个阶段是在到2020年全面建成小康社会的基础上,到2025年基本(basically(0.75))实现社会主义现代化。

**Example 5**:

ST-kor: 시 주석 집권 이후 반대 세력에 대한 대대적인 사정 작업과 감시·규제 확대로 정부 정책에 반대하는 목소리가 거의(nearly(1.75)) 사라진 가운데 나온 주장이어서 이목이 집중되고 있다"

TT-en: As opposing views of the government's policy directions have been subject to nationwide inspection, monitoring and regulation, the professor's argument is gathering public attention (zero translation(1.75))

TT-cn: 习主席执政以来,由于对反对势力的大规模整顿工作和扩大监视和管制,反对政府政策的声音几乎 (nearly) 消失,在这种情况下出现的主张备受关注。

**Example 6**:

ST-kor: 15일 포럼 토론 막판 샤오미(小米) AI제품부 지쉬(季旭) 사장 등 중국 기업 패널들은 "삼성은 중요한 협력 파트너" "메모리반도체 등에서 삼성의 더 많은 지원과 협력을 희망한다" "AI 산업은 반도체 의존도가 매우(very(1.25)) 높기 때문에 삼성에 크게 (中韩关系正面词) 의지한다" 등의 덕담을 건넸다

TT-en: zero translation

TT-cn: 在15日论坛讨论最后阶段,小米的人工智能产品部总经理小米季旭等中国企业嘉宾们表示:"三星是重要的合作伙伴","希望在存储器半导体等领域得到三星更多的支持和合作","人工智能因为半导体依存度问,十分(very(1.25))倚重三星" (The grammatical error here is exactly the same as the original text)

**Example 7**:

ST-kor:트럼프 대통령은 "만약 그것(홍콩 국가보안법 제정)이 일어난다면 우리는 그 문제를 매우 (中美关系负面词) 강하게 다룰 것"이라고 경고했다

TT-en: zero-translation

TT-cn:特朗普总统警告说:"如果发生(制定香港《国家保安法》),我们将非常严厉地处理这个问题"

At level-changing point C, intensifier levels downscale in ET while those in CT remain unchanged. There was an obvious downward trend in the translation of intensifiers that collocate with PGR and PIC in the ET corpus. The down-scale of the PIC number is 4.5 times larger than that of the NIC. PIC intensifiers are frequently used in zero translation, and some PIC sentences are completely lost in ET. The level of amplifiers collocate with China's negative group relation (NGR) has also decreased, with approximately 29% of the down-scale occurring in the negative China-ROK relation and 71% in the negative China-US relation. In addition, 24% of US negative topics are down-scaled in the ET. The Booster "very" refers to China and South Korea's interdependence in positive economic cooperation, which is erased by the translators' zero-translation policy in ET(Example 6). A strong US oppression of China is described in Example 7 using the booster "very," and Korean news media choose a literal translation of the policy in CT, but zero translation in ET, thus obscuring the image of the United States as a sanctioner in English.

The data indicate that ET shows a significant down-scale in PIC. However, negative China-US relations, negative China-ROK relations, and the negative image of the United States have downscaled. A notable point is that the positive relationship between China and South Korea downscales in English, but did not change in Chinese. It is evident that the reshaping of group relationship is influenced by the ideology of the news media.

**Example 8**:

ST-kor: 이어 "나는 중한 관계 발전을 매우 (very(1.25)) 중시하고, 문 대통령과 함께 노력해 양국의 전략적 협력 동반자 관계를 더 높은 수준으로 끌어올리길 바란다"고 덧붙였다

TT-en: In his message, Xi valued the relationship between Beijing and Seoul highly(1.75), hoping to work with President Moon to take their cooperative partnership to a higher level

TT-cn: 他还表示:"我高度(highly(1.75)重视中韩关系发展,愿同文在寅总统一道努力,推动中韩战略合作伙伴关系迈向更高水平"

At level-changing point D, CT upscales while ET remains unchanged where the only one in the Table 3. The Korean news media mostly use literal translation in Chinese, with no apparent level-changing of intensifiers.

At level-changing point E, it is noted that up-scale both appear in CT and ET. In the context of topics describing the relationship between China and South Korea, specifically South Korea's importance to China, both the English translation and the Chinese translation show a significant upward trend. The Booster "매우(very(1.25))" is translated to "highly(1,75)" both in the two target texts.(Example 8)In this way, South Korea is reshaping its importance to China both in the CT and ET.

**Example 9**:

ST-kor: 중학교장회는 "학교를 그만두고 다른 나라로 이민을 선택하는 교사도 7배 이상 증가했다"며 "지난 1년 동안 학생과 교사의 이탈이 상당히(significantly(1.5)) 심각한 상황이라는 것은 명백한 사실"이라고 우려했다

TT-en: The Association of Principals expressed concern that the number of teachers who left school and chose to emigrate to other countries has increased by seven-fold, clearly suggesting the gravity (extreme (2.0) importance or seriousness). of the exodus of both students and teachers for the past year.

TT-cn: 初中校长会表示担忧说:"选择退学移民到其他国家的教师也增加了7倍以上,在过去的一年里,学生和教师的离校现象非常(very(1.25)) 严重,这是不争的事实"

**Example 10**:

ST-kor: 미어샤이머 교수는 "세계는 '2차 냉전'에 돌입하고 있다"며 "중국은 곧 미국과 동등한 힘을 갖게 되고, 앞으로 30년간 경제 성장을 이어간다면 미국을 제치고 가장 (most(1.75)) 강력한 국가가 될 것"이라고 내다봤다 그는 또 "미중이 15년 이내에 대만을 두고 전쟁을 벌일 가능성이 높다고 본다"고 전망했다

TT-en: The world is entering a second Cold War," Mearsheimer said "china will have power on par with the U S soon, and if it continues economic growth over the next 30 years, it will become the most(1.75) powerful country in the world, surpassing the U S There is a strong chance that the U S and China will stage a war over Taiwan within the next 15 years.

TT-cn: zero translation

The CT of intensifiers tends to downscale at level-changing points G, H, and I. Of those, 80% appear in the collocation with NIC and NGR, indicating that the translators intentionally reduce the degree of NIC in CT. It is at the level-changing point H where we see the most obvious manifestation of translators. Intensifiers with NIC show a reverse level-changing direction between ET and CT. In example 9, the Booster "significantly" is translated as Booster "very" with a down-scale in the graduation strength. According to Collins Dictionary, gravity is defined as "the extreme importance or seriousness of a situation or event." Therefore, gravity contains the meaning of the Maximizer "significantly", which is a highly significant level-changing statement in ET. Yet, there are also some downscales of intensifiers that collocate with the PIC in translation, specifically in describing the exaggeration of the Chinese government's power. In example 10, Booster "강력한 (strongly)" collocates with the topics of Chinse power degree which pose a threat to the United States, although the extent of China's strength description is classified as a positive word, from the social context, the up-scale of intensifiers with PIC does not cause a positive image of China.

Thus, in CT, the translator intentionally weakened NIC and NGR in the translation. Moreover, it weakened positive terms related to the power of the Chinese government, so as not to lead Chinese readers to believe that South Korea is not engaged in reshaping the discourse of the China threat theory.

## Discussion

As with previous studies, this study focuses on the study of multilingual translation. This study examines more than just multilingual translation practice [53], but the ideological manipulation behind translated texts. By contrast to previous diachronic studies [13], this study compares the multilingual translations of the same source texts by the same news media at the same period to analyze how ideology affects translations during a specific time period. Additionally, news discourse has been chosen as the research object because its ideological factors are more apparent than those found in literary discourse.

The results obtained from the statistical analysis indicate that the translators use different strategies in translating the intensifiers into the two languages, i.e., CT and ET, which is closely related to the manipulation of the text by the translators.

Firstly, there is a statistically significant difference between the distribution of translation shifts at the 3 levels between ET and CT, with the ET aligning more closely with van Dijk's ideological square model. In terms of topic selection and translation strategies, Korean news media have demonstrated a clear translation propensity of negative strengthening and positive weakening of intensifiers when collocating with China's group relations and China's image words in ET. Cases have been found that violate the model at the lexical level, but the model

still holds when the social context is considered. News media deliberately manipulate up-scale of intensifiers collocating with PIC and PGR in ET. For example, most level-changing cases are designed to deepen the "China threat theory", such as the power of the Chinese government and the goal of China's development, which in essence does not lead to the up-scale of China's positive image. Translation shifts are generally demeaning to others or detrimental to the interests of their own countries.

Second, compared to ET, CT has largely defied Van Dijk's ideological square model. The results indicate that in the translation of a target language, the translation propensity for relevant topics of the target country in which the language is spoken tends to be positive. In most cases, the ET of intensifiers conforms to the ideological square model in which the translation propensity of others is negative. However, the translation propensity of intensifiers in CT is positive when collocating with topics relating to China, "an other" to Korea, which is contrary to the model. In data, there has been a down-scale toward intensifiers in CT, with 80% of the down-scale strategies occurring in the collocation with NIC and NGR, indicating that the South Korean news media has factors to consider in their acceptance of Chinese readers. In addition, the down-scale of intensifiers collocating with PIC and PGR is also associated with the China threat theory, which weakens the degree to which Chinese readers accept negative images of China in target texts. Furthermore, English does not only perform the function of the lingua franca but also the first language of many English-speaking countries, for example, the US. Therefore, we found that negative national relations and the national image of the US are significantly reduced in ET.

In conclusion, the level-changing patterns made by Korean news media about topics of China in ET are more closely aligned with van Dijk's ideological square model than in CT. As a result of the analysis, it can be concluded that the CT of Korean news media about topics of China contains the consideration of Chinese readers, especially for political and diplomatic purposes. When ET of intensifiers collocating with the topics of China, on the other hand, conforms more to the ideological framework, which serves primarily for evaluation purposes. Therefore, the reproduction of the topics about "a target country" in their "target language" is greatly influenced by the target readers, which entails its own set of social, political, and diplomatic factors. This is the particular characteristic of the ideological square model when it is applied to translation.

## Conclusion

The analysis conducted within the framework of the Ideological Square Model reveals significant differences in the translation of intensifiers in Korean news discourse between ET and CT. In contrast to monolingual text, translators do not strictly adhere to van Dijk's ideological square model in translation. Considering the target reader, translators' understanding of the "Us" or ingroup can widen, and the target country that speaks the target language may also be included within the ingroup. Furthermore, group relations of the target country in target texts can be manipulated in line with international relations and diplomatic policies taken by the news media.

The findings indicate that there is a difference in the ideological square model between the monolingual text and multilingual texts. The analysis of the differences in multilingual texts made by the news media indicates that some manipulation and reshaping has occurred. In translation studies, discourse analysis is of importance since it analyzes the linguistic elements of translation And these linguistic elements play a crucial role in shaping the ideological and discursive dimensions of translation texts. By investigating the translation shifts, researchers gain insights into how language and translation practices contribute to the construction and

negotiation of social identities, power dynamics, and ideological perspectives in cross-cultural communication.

Moving forward, besides national image and international relations, studies of multilingual translation texts can also examine ideological factors such as feminism, racism, power relation, etc. Additionally, examining the role of multilingual translation texts in the construction of narratives could provide further insights into ideological dynamics in this increasingly complex world.

## Supporting information

**S1 File.**
(ZIP)

## Acknowledgments

I would like to express my heartfelt gratitude to Associate Professor Li Tao from Shanghai Ocean University for his invaluable guidance and support throughout the development of this research paper. His expertise and insights were instrumental in shaping the theoretical framework, research methodologies, and writing of this paper.

His encouragement have been pivotal in my academic journey, and I am deeply appreciative of his unwavering dedication to my growth as a researcher.

I would also like to extend my thanks to Tu Haoyan, from the Quality Assurance Department of Changchun Keygen Biological Products Co., Ltd. His valuable insights and guidance on statistical analysis methods were instrumental in the completion of this research.

Furthermore, I would like to extend our sincere gratitude to the editor and reviewers of this journal. Their expert guidance and valuable insights were instrumental in enhancing the quality of my research. Their meticulous review and constructive feedback helped me refine the content and structure of the paper, making it more rigorous and persuasive.

In the end, I would like to express my appreciation to Professor Zou Leilei of the Foreign Language Department at Shanghai Ocean University for her invaluable guidance and support during my graduate studies. Her inspiration has significantly shaped my scholarly perspective and instilled in me a greater sense of rigor in academic pursuits.

## Author Contributions

**Data curation:** Quan Jiuding.

**Formal analysis:** Quan Jiuding.

**Methodology:** Quan Jiuding.

**Writing – original draft:** Quan Jiuding.

**Writing – review & editing:** Quan Jiuding.

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
