## [Decision Letter · Decision Letter 0]

9 Aug 2023

PONE-D-23-20280Bilingual Translations of Intensifiers in Dong-A Ilbo’s News about   China： A  Corpus-based Discourse A nalysis ApproachPLOS ONE

Dear Dr. Quan,

Thank you for submitting your manuscript to PLOS ONE. After careful consideration, we feel that it has merit but does not fully meet PLOS ONE’s publication criteria as it currently stands. Therefore, we invite you to submit a revised version of the manuscript that addresses the points raised during the review process.

We look forward to receiving your revised manuscript.

Kind regards,

Anastassia Zabrodskaja, Ph.D.

Academic Editor

PLOS ONE

Journal Requirements:

5. Please ensure that you include a title page within your main document. You should list all authors and all affiliations as per our author instructions and clearly indicate the corresponding author.

7. We note you have included a table to which you do not refer in the text of your manuscript. Please ensure that you refer to Table 4 and 5 in your text; if accepted, production will need this reference to link the reader to the Table.

Additional Editor Comments:

Please check carefully what the reviewers are telling you and modify accordingly.

Reviewers' comments:

Reviewer's Responses to Questions

**Comments to the Author**

1. Is the manuscript technically sound, and do the data support the conclusions?

Reviewer #1: Partly

Reviewer #2: Partly

2. Has the statistical analysis been performed appropriately and rigorously? 

Reviewer #1: No

Reviewer #2: I Don't Know

3. Have the authors made all data underlying the findings in their manuscript fully available?

Reviewer #1: Yes

Reviewer #2: No

4. Is the manuscript presented in an intelligible fashion and written in standard English?

Reviewer #1: No

Reviewer #2: Yes

5. Review Comments to the Author

Reviewer #1: The manuscript (PONE-D-23-20280) addresses an interesting question of the function of ideology in translation of the source text into more than one target language, particularly with respect to the translation of news discourse related to diplomatic relations between South Korea, China, and the United States. However, there are some weaknesses with the research methodology, theoretical analysis, language, and style.

1. Research methodology

Although the research question and the theoretical framework of the paper are clear, we do not know how the author uses the framework to deal with the question. The relevant literature is not provided and we do not know the contribution of this study to the existing knowledge of translation studies. Some key terms in the paper such as “ideology”, “ideological square”, “translation shift”, “intensifier”, “downtoner”, “transediting”, “translation strategy”, and “fidelity” are not explicitly defined. The terms “literal translation”, “free translation”, and “zero translation” used in this paper are not in their universally acknowledged sense in translation studies. There is no solid basis for the judgment of the fidelity of intensifiers in Table 2. The structure of the RESULTS part is somewhat illogical, particularly with respect to the HYPOTHESES section.

2. Theoretical analysis

Who are the actors such as publishers, editors, and translators in the translation of Korean news into English and Chinese? They play a crucial role in production of translation shifts. Unfortunately, the author fails to make an analysis of the reasons why they make those shifts or employ the strategies of “free translation” and “zero translation”. The actor network theory may be combined with the ideological square theory to explore the issue. If possible, the actors or translators can be interviewed or questionnaired in order to collect the relevant data or evidence. The news censorship or gatekeeping mechanism in news production may be an important dimension in such explorations.

3. Ungrounded pronouncements

There are a number of claims in the paper which are unsupported by any citation or evidence, such as “Today, multilingual translation texts are most often found in the news media due to globalization”, “In the digital era, news discourse interpretation plays an increasingly significant role in constructing ideologies for global audiences”, and so on.

4. Language and style

There are a large number of linguistic and stylistic infelicities in the paper such as grammatical mistakes, typos, and terminological inconsistencies. For example, in translation studies, the term “target text” is used to refer to the text to be translated. However, the author uses such terms as “translation text”, “targeted text”, and “translated text”. There are many abbreviations used in the paper, such as “ET” and “CT”. The author fails to provide the full name for some of the abbreviations when they appear in the paper for the first time. A few tables are not well designed, such as Tables 2 and 3 where the information on Chinese and English translations is confusing. The language in a few English and Chinese translations in the paper is problematic, as illustrated by “人工智能因为半导体依存度问” in example 6.

Reviewer #2: The topic sounds interesting and engaging for the readers. However, the data analysis method is very old. The paper has a few issues that I am listing down here for your information:

1. The discussion of Translation Strategies is a very old and outdated matter that nowadays are good for student assignments in the classroom than to be appeared in scientific journal articles.

2. There is no comprehensive Literature Review to show what the previous studies in the field have achieved and where the gap is. The author needs to make sure that he/she covered the major and leading studies in the field in the LR section to justify the research gap.

3. There is no link between the LR and the methodology and data analysis. There should be a novel flow in the article in order to provide inter-related information for the readers throughout the paper.

4. All the examples in foreign languages must be accompanied by an English gloss translation.

5. The research is reporting some numbers calculated by the author with no/little justification on the design and method.

6. There is no justification for bringing a handful of examples from the big corpora to show the ideological differences. It makes the data bias.

7. There is no major/minor contribution to the field of Translation Studies.

My advice to the author is to read the major studies in the field to have a clear understanding of the topic before jumping to analysis and conclusion.

6. PLOS authors have the option to publish the peer review history of their article (what does this mean?). If published, this will include your full peer review and any attached files.

Reviewer #1: **Yes: **Chuanmao Tian

Reviewer #2: No

---

## [Author Response · Author response to Decision Letter 0]

12 Aug 2023

I have finished revising the manuscript and uploaded all of my corpus data and related websites.

---

## [Decision Letter · Decision Letter 1]

4 Sep 2023

PONE-D-23-20280R1Bilingual Translations of Intensifiers in Dong-A Ilbo’s News about China：A Corpus-based Discourse Analysis ApproachPLOS ONE

Dear Dr. Quan,

Thank you for submitting your manuscript to PLOS ONE. After careful consideration, we feel that it has merit but does not fully meet PLOS ONE’s publication criteria as it currently stands. Therefore, we invite you to submit a revised version of the manuscript that addresses the points raised during the review process.

ACADEMIC EDITOR: Please make the appropriate changes.

We look forward to receiving your revised manuscript.

Kind regards,

Anastassia Zabrodskaja, Ph.D.

Academic Editor

PLOS ONE

Reviewers' comments:

Reviewer's Responses to Questions

**Comments to the Author**

1. If the authors have adequately addressed your comments raised in a previous round of review and you feel that this manuscript is now acceptable for publication, you may indicate that here to bypass the “Comments to the Author” section, enter your conflict of interest statement in the “Confidential to Editor” section, and submit your "Accept" recommendation.

Reviewer #1: All comments have been addressed

2. Is the manuscript technically sound, and do the data support the conclusions?

Reviewer #1: Partly

3. Has the statistical analysis been performed appropriately and rigorously? 

Reviewer #1: No

4. Have the authors made all data underlying the findings in their manuscript fully available?

Reviewer #1: Yes

5. Is the manuscript presented in an intelligible fashion and written in standard English?

Reviewer #1: No

6. Review Comments to the Author

Reviewer #1: The revised manuscript (PONE-D-23-20280_R1) has just added a few sentences in the METHODOLOGY part to clarify a few issues on the data collection, and thus failed to make a substantial improvement according to the reviewers’ comments. If I were the author, I would carefully revise the paper and incorporate the reviewers’ constructive suggestions into the revision.

7. PLOS authors have the option to publish the peer review history of their article (what does this mean?). If published, this will include your full peer review and any attached files.

Reviewer #1: **Yes: **Chuanmao Tian

---

## [Author Response · Author response to Decision Letter 1]

8 Sep 2023

I would like to extend my sincere gratitude to the reviewers for their valuable feedback and insightful comments on my manuscript. Your thoughtful reviews have significantly contributed to the improvement of this research.

I also want to express my apologies for not addressing some of the suggestions made in the previous review comprehensively. Your feedback is highly appreciated, and I have taken it to heart. I have now revisited the manuscript with great care, making the necessary revisions and enhancements to ensure its quality and clarity.

Your constructive input has been invaluable, and I am truly thankful for your dedication to the peer-review process. Your efforts have undoubtedly strengthened the overall quality of this work. Once again, I appreciate your time and expertise in reviewing my manuscript.

---

## [Decision Letter · Decision Letter 2]

14 Sep 2023

PONE-D-23-20280R2Bilingual Translations of Intensifiers in Dong-A Ilbo’s News about China：A Corpus-based Discourse Analysis ApproachPLOS ONE

Dear Dr. Quan,

Thank you for submitting your manuscript to PLOS ONE. After careful consideration, we feel that it has merit but does not fully meet PLOS ONE’s publication criteria as it currently stands. Therefore, we invite you to submit a revised version of the manuscript that addresses the points raised during the review process.

We look forward to receiving your revised manuscript.

Kind regards,

Anastassia Zabrodskaja, Ph.D.

Academic Editor

PLOS ONE

Journal Requirements:

Additional Editor Comments:

Please implement these minor changes.

Reviewers' comments:

Reviewer's Responses to Questions

**Comments to the Author**

1. If the authors have adequately addressed your comments raised in a previous round of review and you feel that this manuscript is now acceptable for publication, you may indicate that here to bypass the “Comments to the Author” section, enter your conflict of interest statement in the “Confidential to Editor” section, and submit your "Accept" recommendation.

Reviewer #1: All comments have been addressed

2. Is the manuscript technically sound, and do the data support the conclusions?

Reviewer #1: Yes

3. Has the statistical analysis been performed appropriately and rigorously? 

Reviewer #1: Yes

4. Have the authors made all data underlying the findings in their manuscript fully available?

Reviewer #1: Yes

5. Is the manuscript presented in an intelligible fashion and written in standard English?

Reviewer #1: Yes

6. Review Comments to the Author

Reviewer #1: The manuscript (PONE-D-23-20280R2) has addressed most concerns raised in the reviewer's comments, and the revisions are basically acceptable. It is suggested that some of the author's responses can be incorporated into the article, such as Response 6.

7. PLOS authors have the option to publish the peer review history of their article (what does this mean?). If published, this will include your full peer review and any attached files.

Reviewer #1: **Yes: **Chuanmao Tian

---

## [Author Response · Author response to Decision Letter 2]

20 Sep 2023

I have incorporated explanations regarding the translator's definition and potential misunderstandings that could arise in Example 6.

---

## [Decision Letter · Decision Letter 3]

25 Sep 2023

Bilingual Translations of Intensifiers in Dong-A Ilbo’s News about China：A Corpus-based Discourse Analysis Approach

PONE-D-23-20280R3

Dear Dr. Quan,

We’re pleased to inform you that your manuscript has been judged scientifically suitable for publication and will be formally accepted for publication once it meets all outstanding technical requirements.

Kind regards,

Anastassia Zabrodskaja, Ph.D.

Academic Editor

PLOS ONE

Additional Editor Comments (optional):

Reviewers' comments:

Reviewer's Responses to Questions

**Comments to the Author**

1. If the authors have adequately addressed your comments raised in a previous round of review and you feel that this manuscript is now acceptable for publication, you may indicate that here to bypass the “Comments to the Author” section, enter your conflict of interest statement in the “Confidential to Editor” section, and submit your "Accept" recommendation.

Reviewer #1: All comments have been addressed

2. Is the manuscript technically sound, and do the data support the conclusions?

Reviewer #1: Yes

3. Has the statistical analysis been performed appropriately and rigorously? 

Reviewer #1: Yes

4. Have the authors made all data underlying the findings in their manuscript fully available?

Reviewer #1: Yes

5. Is the manuscript presented in an intelligible fashion and written in standard English?

Reviewer #1: No

6. Review Comments to the Author

Reviewer #1: The manuscript (PONE-D-23-20280_R3）has basically addressed the questions raised in the comments. Thank you! However, the language and style of the article can be improved, such as in-text citation format (the abbreviations of the scholars' given names need not be provided; the full names (or the surnames) of the scholars should be provided in their normal order in the text proper), and grammatical infelicities (e.g. "The grammatical error here is exactly the same as the original text").

7. PLOS authors have the option to publish the peer review history of their article (what does this mean?). If published, this will include your full peer review and any attached files.

Reviewer #1: **Yes: **Chuanmao Tian

---

## [Editor Report · Acceptance letter]

28 Sep 2023

PONE-D-23-20280R3 

Bilingual Translations of Intensifiers in Dong-A Ilbo’s News about China：A Corpus-based Discourse Analysis Approach 

Dear Dr. Jiuding:

I'm pleased to inform you that your manuscript has been deemed suitable for publication in PLOS ONE. Congratulations! Your manuscript is now with our production department. 

Kind regards, 

on behalf of

Professor Anastassia Zabrodskaja 

Academic Editor

PLOS ONE